# A Case Study of Facial Emotion Classification Using Affdex

**DOI:** 10.3390/s19092140

**Published:** 2019-05-09

**Authors:** Martin Magdin, Ľubomír Benko, Štefan Koprda

**Affiliations:** Department of Informatics, Constantine the Philosopher University in Nitra, Tr. A. Hlinku 1, 949 74 Nitra, Slovakia; lbenko@ukf.sk (L.B.); skoprda@ukf.sk (Š.K.)

**Keywords:** face analysis system, emotions, valence, facial expression analysis

## Abstract

This paper focuses on the analysis of reactions captured by the face analysis system. The experiment was conducted on a sample of 50 university students. Each student was shown 100 random images and the student´s reaction to every image was recorded. The recorded reactions were subsequently compared to the reaction of the image that was expected. The results of the experiment have shown several imperfections of the face analysis system. The system has difficulties classifying expressions and cannot detect and identify inner emotions that a person may experience when shown the image. Face analysis systems can only detect emotions that are expressed externally on a face by physiological changes in certain parts of the face.

## 1. Introduction

Nowadays, face analysis systems are widely used in various areas of life and the educational process is one of them. Face analysis systems are commonly used by researchers to help eliminate the problematic parts of the learning process, to understand the learner’s inner state of mind and to help the learner overcome potential stressful conditions of their studies [1,2]. Face tracking, an extraction of a specific area of interest and subsequent classification, must be run in real time so that researchers may work with the actual data. This way, it is possible to obtain relevant results, and with feedback, it is possible to interfere with the course of the educational process. The findings may be applied and any errors, which might be indicated by negative feelings resulting in an emotional state of the learner removed.

The localization methods and techniques, subsequent face detection in space, the extraction of a specific area of interest and the classification of emotional states have made significant progress in recent years. At present, these techniques are successfully applied in various devices, such as digital cameras, or software applications, e.g., facial recognition on the Facebook social network. However, current classifiers have a drawback. The classifiers learn based upon instructed and thus artificial displays of emotion in the majority of cases. There were many databases created in the past where such expressions are included (such as Jaffe, Caltech Faces 1999 Database, BaoDataBase, YALE). As it was shown in the last five years [3,4], this approach was incorrect. Instructed facial expressions, which were meant to characterize a given emotional state, represent an exaggerated situation (an unrealistically high level of a given emotion), and thus the resulting success rate of these classifiers has reached ca. 90%. Due to this reason, additional techniques for obtaining facial datasets were created. These would reflect genuine emotional states [5]. Despite all that, it is still not clear, whether current systems can capture real feelings and classify specific emotional states. According to Ekman, classification instead of capturing only exaggerates expressions from escalated situations.

An experiment carried out using Affectiva’s Affdex SDK is presented in this paper. This SDK is one of the most widely used face analysis systems. The aim of the experiment is to analyze face analysis systems with a focus on evaluating the results of the system compared to the expected student responses.

The rest of the paper is structured as follows. Section 2 contains the survey of existing face analysis systems in the context of their historical development. This point of view is crucial because both the determination of the emotional state and the development of face analysis systems took place simultaneously. Section 3 describes the methodology and the motivation for the experiment as well as the input conditions and the hypothesis of the experiment. Section 4 deals with the results of the experiment and its analysis. Furthermore, discussion and conclusions are offered in the last section.

## 2. Related Work

The issue of recognition and determination of human emotions has been an important research area ever since the time of Charles Darwin who was the first one to point out the link between evoked emotional states and the characteristic expressions of the human face. Darwin’s theory has been confirmed by several different studies over subsequent decades [6,7,8,9,10,11]. Darwin assumed that emotional expressions are multimodal behavioral patterns of an individual, and thus formed his own detailed descriptions of more than 40 emotional states [12]. Over the last century, several different models for emotion classifications, ranging from universally experienced basic emotions to unique and complex ones were psychologically defined. Two of the models researched in the field of emotion recognition [13,14,15,16] have been predominantly used in the last decade: the basic classification of six emotional states by Ekman [17] and Russel’s circumplex model of emotions [18].

Contrary to Ekman’s classification, Russel’s model is not as strictly separated and indicates that the emotional states of an individual are dynamic multimodal patterns of behavior. For example, the expression of fear on the face consists of the widening of the pupils while contracting the muscles around the mouth. On the other hand, the expression of joy on the face consists of the reduction in pupil movement and concurrent substantial change in the shape of muscles around the mouth. Russel’s circumplex model assumes that under certain conditions, there may be an overlap of some features that could uniquely classify a given type of emotion (for example, happiness and surprise; fear and sadness, etc.). However, many authors have recently indicated that in order to classify various emotional states, it is necessary to recognize the fact that emotions allow such expressions primarily through changes in physiological processes [19,20]. Various approaches [21,22] that were able to detect responses to relevant conditions of individuals, such as behavior, physiological and empirical components [23] were proposed due to these changes.

Current face analysis systems, which are able to determine the emotional state of an individual from the facial expressions analysis, operate in three basic phases, as defined by Kanade [24]:Face detection phase,Feature extraction,Classification of emotions according to the selected model.

Interest in this area dates to the 1960s when Bledsoe, Helen Chan, and Charles Bisson created the first face recognition algorithm [25,26,27,28]. Their approach and technique were later used by Goldstein, Harmon, and Lesk for the facial feature extraction phase [29].

The first fully-functional system was implemented by Kanade in 1973 [24]. The algorithm was able to automatically evaluate 16 different facial parameters by comparing extractions obtained using the computer with extractions delivered by a man. However, the system performed a correct identification with a success rate in the range of only 45%–75%.

Over the years, several detection methods emerged that also could have been applied in the next stages of the recognition process. In 2002, Yang introduced a classification [30], mostly used by many other authors, consisting of knowledge-based methods, feature invariant approaches, template matching methods, and appearance-based methods.

A number of different algorithms for knowledge methods were proposed by Yang and Huang [31]; Kotropoulos and Pitas [32]; Zhang and Lenders [33]. The research of feature invariant approaches was carried out by Vezhnevets et al. [34]. Lakshmi and Kulkarni [35] used skin color to improve detection accuracy in combination with the grayscale edge of the input image. Ghimire and Lee [36] and Chavhan et al. [37] proposed a new algorithm that used improved image input (by use of histogram or snap technique) in the pre-processing of the image and a combination of skin color and edge information to improve face detection rate, as well as verifying individual candidates - feature points on the face (nose, eyes, mouth).

Among the oldest methods based on template matching is the algorithm proposed by Sakai et al. [38]. This algorithm used several sub-templates for the eyes, nose, mouth, and face to create an exact face model. Later, different templates using predefined patterns were designed by researchers [39,40,41,42]. Wang and Tan [43] suggested a procedure using an ellipse as a template for describing the face and its parts that proved to be ineffective over time due to different dimensions of a face, gaze fixations or a potential face rotation during the detection.

Appearance-based methods are derived from template matching methods but to identify and recognize individual interest areas, a statistical analysis or machine learning (support vector machine), which is indicative of the extraction and classification phases is used. Given a large amount of data that is necessary to be processed for these methods, a common approach is to reduce the dimensions of the detected area (dimensionality reduction), thereby achieving higher computational efficiency and higher success rate of the detection itself. Among these methods, the most prominent techniques are AdaBoost (Viola-Jones detector) algorithm, S-AdaBoost algorithm [44], FloatBoost algorithm [45], hidden mark model, Bayes classification, support vector machines (SVM), and neural networks. 

The correct division of basic extraction methods is not uniform to this day. Most of them are used in areas other than face recognition. All these techniques may be used either in the extraction phase or in the classification phase. At the beginning of the 1990s, the first division of these methods appeared using these algorithms and techniques [46,47]. This division contained five basic techniques:principal component analysis,neural network-based methods,active shape modeling,active appearance model,Gabor wavelet.

A considerate amount of research was conducted on the application of Gabor wavelets, or Gabor filters [48,49]. Coefficients are the output of using the Gabor filters, and they are used to describe the emotion by changes of the individual facial features, for example, the change in the position of the eyebrows, mouth, cheekbones, etc. Application of these coefficients results in the creation of multi-point masks [50] consisting of action points. This action points form together a system called the geometric model. However, these geometric models are expressions of an ideal case of a human face.

In the case of the active shape model (ASM), the feature points are constantly compared with the real physical state of an individual and thus describe the actual areas as closely as possible. ASM was first used in 1988 [51] and later, this model, based on the so-called "snakes" (generically active facial features), was improved by Yuille [52], Wang [53], and Yoo [54].

The classification of emotions is the final phase of the automatic analysis of the emotions from extracted features. However, the algorithms used in the classification are commonly applied in the previous two phases as well. 

A typical example is the hidden Markov model (HMM), which represents the statistical model, the formal basis for the creation of probabilistic models. HMM is most commonly represented as a simple dynamic Bayesian network and can be used in all three phases of the recognition process [55].

Rajakumari [56] applied HMM in a classification of the emotional states by measuring the distance between the eyebrow and the iris. They classified the emotional state of anger, fear, happiness, and disgust by measuring the distance.

Similarly to HMM, neural networks may be used in all three phases of the recognition process. An example of the use of neural networks in the detection phase may be the method proposed by Rowley et al. [57]. Training a neural network makes it possible to achieve a very successful classification [58].

Wang et al. [59] proposed a Bayesian scene-prior-based deep learning model focused on extracting features from the background scenes. The authors used Bayesian ideas to train the scene model based on a labeled face dataset. The novel approach transforms a face image into new face images by referring to the given face with the learnt scene dictionary. 

The support vector machine (SVM) method has been used since 1995 in various areas of detection and classification [60,61]. SVM represents a linear classification algorithm which geometrical interpretation can be understood as a search for an ideal hyperplane that separates the attributes of two different categories [62,63].

Despite the many methods and techniques that have been developed over the last 40 years, the current SDK for fully automated face analysis systems are still not immune to the following influences [64]:the facial position and the incorrect angle of the scanning device where certain parts of the face may be out of focus (for example, part of the mouth or nose, eye, etc.),incorrect face and sensor distance causing loss of facial features information,illumination of the scene,smaller obstacles that temporarily cover a certain part of the face (for example, long hair), which can cause a partial loss of the information or act as a disturbing element,accessories and additional features such as glasses, mustache, etc. Since these elements are often permanent features characterizing a face, it is necessary to include these elements in the eventual detection,ethnicity, age, sex, appearance, and other attributes, by which the faces of various people differ.

The statements of Rizvi et al. [64] are also confirmed by Wu et al. [65] who state that race (ethnicity) should be included among the basic and key attributes of facial analysis. The issue with automatic face analysis is that traditional machine learning methods deal with race classification only in combination with two separate steps: extracting artificially designed features and training the right classifier with these features. According to Wu et al. [65], there are other ways to eliminate these issues.

Face recognition in images is one of the most challenging research issues in tracking systems (or also as part of an access system) because of different problems [66]. Among these problems are various non-standard poses or expressions using extracted the facial parts. A simple lighting change can be often enough to lead to an incorrect classification. Thus, the robustness of the recognition method relies heavily on the strength of the extracted functions and the ability to handle both face and low-quality images.

In terms of computational power is the reduction in data and function in the face recognition process essential and the researchers have recently been focused on the use of modern neural networks for automated analysis [67].

Nowadays, the above-mentioned impacts are dealt with by novel approaches to increase the overall classification success. Therefore, facial expression recognition (FER) is important for the transition from the instructed (stimuli) or laboratory-controlled expression to real face expression. The deep learning techniques (deep neural networks) removed various problems as illumination, head pose and identity bias. The ability to learn robust features from the raw face image makes deep convolutional neural networks (DCNNs) attractive for face recognition.

The usage of deep learning is currently really wide. Despite the fact that it is possible to gather a great amount of data and information from sensors, the data are often not possible to process or use. Rapidly developing areas of deep learning and predictive analysis have started to play a key role in healthcare development and research to understand patients’ feelings and needs if they are bedridden and are unable to communicate with the doctor in a standard way [68,69]. 

In this way, deep learning can be successfully used, for example, for the quantitative analysis of polio. Polio, also known as Bell’s palsy, is the most common type of facial polio. The facial polio causes loss of muscle control in the affected areas, which represents not only face deformity but especially facial expression dysfunction and thus the inability of most current algorithms to capture the patient’s true emotions. According to the findings of Hsu et al. [70], current approaches to the automatic detection of polio in childhood take into account in the most cases manual functions, resulting in incorrect classification of patients’ emotional status.

Wang et al. [71] apply a completely novel approach that is based not only on face detection but also on sensors. The authors created an emotion-based algorithm based on eye movement information by collecting and analyzing eye-tracking and eye movement sensing. Specifically, the authors extracted time-frequency motion functions of the eyes by first applying a short-term Fourier transformation to raw multi-channel eye-tracking signals. Consequently, in order to integrate time-domain movements in time (i.e., saccade duration, fixation duration and pupil diameter), two fusion function strategies were investigated: function-level fusion (FLF) and decision-level fusion (DLF). Recognition experiments were also performed according to three emotional states: positive, neutral, and negative. The average accuracy, however, is 88.64% (FLF method) and 88.35% (DLF method).

Therefore, current methods in this area focus on deep learning and so-called emotional computing. A study focused on emotional computing [72] is one of them. Results of the research are currently used in home appliances or the automotive industry [73]. Chaitchotchuang et al. [74] used Markov chains to classify only two emotional states (negative/positive). This way, the authors created an emotional probability that simulates a dynamic process of spontaneous transfer of an emotional state. This method offers a novel approach to study the emotional state classification such as emotional calculation and the emotion automation generation theory. According to Kyung-Ah et al. [75], the introduction of the mathematical model is the key feature of human-computer interaction to the current SDK focused on face analysis and classification of emotional states. The aim of the mathematical model is to help the psychologists better understand, determine correctly, and express the essence of natural emotion, especially in the decision-making process.

Out of the many available mathematical models, apart from the Markov chains, the deep learning model (DLM) is currently the most interesting. It is mainly due to the model’s ability to overcome the disadvantages of traditional algorithms [72]. Deep learning offers the most powerful option of machine learning in the case of hierarchical models which are used in many aspects of our lives (speech recognition, pattern recognition or computer vision). In principle, deep learning models are designed as convolutional neural networks (CNNs) that are mainly used in the second and the third phases of the recognition process, for the extraction and subsequent classification. The new CNN architecture for facial expression recognition was proposed by Kuo et al. [76]. Their solution in the frame-to-sequence approach successfully exploits temporal information and it improves the accuracies on the public benchmarking databases.

In the case of the extraction (for example, restricted Boltzmann machines), DLM offers a reduction of computational complexity and overall acceleration from the point of computational view while maintaining sufficiently good input results for the classification [77]. Thus, restricted Boltzmann machines and CNN are becoming powerful pre-coding techniques in deep learning for the use of classification [78,79,80]. In the case of using the deep learning methods, such as the deep belief network, it is possible to discover a very small error in the classification of emotional states (1.14%–1.96%) [72]. The issue with DLM that are a part of the current SDK is, despite the small error rate in classification, the inability to capture and correctly classify the change of emotional state, for example, in the decision-making process of a person if the reaction time is very short [3].

However, Bazrafkan et al. [81] showed that the issue of deep neural networks (DNN) in the classification phase is the accuracy that is significantly lower when the network is trained with one database and tested with another database.

Bahreini et al. [82] are developing a novel software for a correct recognition and classification of emotional states. The software offers to recognize the emotion from the image files or uploaded video files. It can use real-time data from a webcamera and classify the so-called subtle emotional facial expressions. It uses the FURIA algorithm for unordered induction of fuzzy rules. This algorithm makes it possible to timely detect and return appropriate feedback based on the facial expressions. The success rate of the algorithm is 83.2%.

Shan Li and Weihong Deng [83] deal with the major issue of the FER in classical categorical model (e.g., Ekman) that widely uses the definition of the prototypical expressions. This definition includes only very small portion of specific categories and cannot mark the real problems of expressive human behaviors in real time interactions.

According to Samadiani et al. [84], although laboratory FERs achieve very high accuracy (approximately 97%), the issue is the technical transfer from the lab to real application where the authors face a large barrier with very low accuracy (around 50%).

Based on the results of Rizvi et al. [64], the aim of the introduced research was to verify the reliability of the most commonly used SDK—Affectiva’s Affdex SDK. Other frequently used face analysis systems operate on the similar principle (e.g., FACET, Eyeris EmoVu, InSight SDK). Based on this, for the aim of the experiment was chosen the Affdex module. Also, the Affdex module offers its application for testing in the academic environment.

## 3. Materials and Methods

The aim of the experiment was to analyze the face analysis system with a focus on evaluating the results of the system in comparison to the expected student responses.

In most cases, two basic ways of performing face analysis for the learning process are used:using a camera to capture all the students’ faces at the same time,using multiple cameras, each capturing only one student’s face at the time.

The first approach is highly dependent on a computational performance. The second approach is also dependent on the amount of hardware used, as each student whose face would be analyzed would have to have their own separate camera.

The experiment was designed with the intention of identifying and removing errors in the learning process while understanding students’ inner state of mind during the learning process or helping them overcome stressful learning conditions. The performed experiment should confirm or invalidate the eligibility of the face analysis system in the learning process. Based on this, the following hypotheses were established:It is assumed that emotional facial expressions (facial emotions) can be correctly identified and classified using the Ekman’s classification and the selected face analysis system.It is assumed that the subconscious reaction to the positive or negative emotional states correlates with the facial expressions (facial emotions) that are recognizable by the face analysis system.

In the experiment, 50 participants were examined, representing a standard sample of first-year university students at the age 18–19 years who agreed to participate in the experiment during the term. The students have given consent to scan their faces before performing the experiment. The consent was in accordance with the current GDPR standards. The very faces of the students were not important for the experiment, so they were not stored.

There were a 100 random images shown to every student. Then, their reactions to the image were recorded. Images changed automatically with a timeout of five seconds. From the psychological point of view, images were meant to induce positive or negative emotional states [85]. The NAPS standardized image database [86] was used for this experiment where the images were rated (positive, negative or neutral rate for each image). Emotional facial expressions identified by the selected face analysis software Affdex should correlate with the subconscious response of the participant to the image displayed.

The selected software Affdex by Affectiva (in collaboration with iMotions) is currently one of the most used SDKs for classifying emotional states. Affdex uses the facial action coding system (FACS), a complex and widely used objective taxonomy for coding facial behavior. The Affdex system for automated facial coding has four main components [87]:face and facial landmark detection,face texture feature extraction,facial action classification,emotion expression modeling.

A histogram of oriented gradient (HOG) is used to process the input (captured) camera feed. This step also extracts the needed areas of interest. A SVM trained on 10,000 images of faces from all over the world is applied to the input, which determines a score of 0–100. The classification of the emotional state itself is realized according to Ekman’s classification. However, Affdex includes an additional state—contempt. The classification uses the EMFACS standard coding system [88], based on the original Ekman coding system FACS [89], with emotional expressions getting a score of 0–100, where 0 is a missing expression, 100 is a fully-present expression.

The valence parameter (pleasantness of the emotion) has been used to determine whether a subconscious reaction in relation to a positive or negative emotional state correlates with the emotional expressions in the face that are recognizable by the Affdex face analysis system. Valence is used in emotions to determine the positive or negative emotional state. It is not an emotion but a component of emotion that represents the subjective pleasantness of a psychological state.

The use of valence makes it possible to characterize and categorize specific emotional states, such as anger and disgust (negative valence) or joy, surprise (positive valence) [5]. Images from the NAPS standardized database were assigned appropriate valence by an expert (−1 for the negative state, 0 for the neutral state, 1 for the positive state), resulting in the specified valence. The Affdex system also includes the ability to measure valence.

## 4. Results

The experiment was carried out on a sample of 50 first-year university students. Every student was gradually shown 100 images from a database consisting of a total of 899 images from a standardized NAPS database. Related images were categorized for better evaluation. There were 93 categories of images (such as a lake, sport, building, injury, fire, boredom, depression, etc.). Each image was assigned its specified emotion valence– positive (1), neutral (0) or negative (−1). Images were displayed at random order and the students’ responses (emotions) were recorded using Affdex software. Then, values for seven emotions were recorded in the log file: joy, fear, disgust, sadness, anger, surprise, and contempt. At the same time, a student’s valence value for the given image was recorded, which represented a positive, neutral or negative emotional state response to the shown image. The emotion values ranged from 0–100, representing a percentage (the sum of all recorded emotions and neutral emotion was 100%). Valence ranged from −100 to 100, where the positive value represented a positive emotional state response, the negative value represented a negative emotional state response, and zero represented a neutral emotional response. After obtaining the data, the preparation of data consisted mainly of the transformation of variables (categorization of valence variable and transformation of variables representing individual emotions) and removing the measurements recorded before displaying the first image. The log file contained 5547 records after data pre-processing. Each record represented a recorded student’s response for a specific image. 

At first, the recorded response of the emotional state for each image with the specified (expected) image response was compared. Evaluation of the obtained data was performed by statistical processing using the Chi-square test and the contingency coefficients. The only assumption of validity of the Chi-square test is that the expected counts are greater than or equal to five. This condition was met in all cases. The degree of statistical dependence between qualitative features was judged based on the contingency coefficient *C*, and Cramér’s *V*. Values close to zero indicate a weak dependence and values approaching one represent a strong dependence [90]. In the case of specified valence and observed valence, a trivial degree of dependence was identified, and statistically significant differences were not found between individual emotions (Table 1).

Based upon these findings, it can be said that using Affdex software, it wasn’t possible to recognize correct emotional states from facial expressions. By comparing the observed valence value with a specified valence, a match of only 21.64% cases was observed. In most cases, a positive reaction was expected (almost 58% of displayed images were with predicted positive reaction). On the contrary, the least number of displayed images had a predicted neutral response (about 17%). As shown in the plot (Figure 1), for each predicted emotion a different reaction was recorded by Affdex. In the case of predicted positive and negative reactions of displayed images, a neutral response prevailed. Only in the case of predicted neutral reactions, the recorded reaction matched the prediction. The observed students showed mainly a neutral reaction (in almost 70% of cases), in contrast, the least observed was a positive reaction (only 7% of cases). The negative reaction showed a relatively close ratio of expected and observed reactions (25% of expected vs. 23% observed). However, in this case, there were negative reactions observed mostly in cases where a positive response was expected. This might have been caused either by the wrong classification of displayed images (images that were expected positive reaction caused a neutral or negative reaction) or by the inaccurate evaluation of emotion expressed by a student.

Since individual images were categorized in advance and in most cases, the categories represented either positive, neutral or negative reactions, it was decided to analyze the connection between categories and recorded reactions of the students. For all three categories and observed responses (valence) a small degree of dependence was identified, but only for neutral response categories, a statistically significant difference was demonstrated at 95% significance (*p* < 0.05) (Table 2, Table 3 and Table 4).

The highest degree of match in the neutral reaction categories was achieved for the “bored” category (Figure 2).

In the case of the category evaluation, only the categories with at least 10 displayed images were taken into account. The “bored” category reached a 77.5% success rate, so the expected neutral response was seen in 31 out of 40 shown images. As an example of used images we chose an image from this category, for which the recorded reactions were neutral in all six of its showings, as was the expected reaction (Figure 3).

The next step was a comparison of recorded and expected individual emotions. Based on a simple separation to positive and negative emotions, a new variable was created. This variable was transformed to either positive, neutral or negative reaction, depending on the recorded emotions. When the share of positive emotions (joy and surprise) prevailed, the response for the image was classified as positive. When the share of negative emotions (fear, disgust, sadness, anger, and contempt) prevailed, the response for the image was classified as negative. If neither positive nor negative emotions prevailed, the response for the image was classified as neutral. Although Affdex does not log neutral responses, the sum of all the emotions equals to 100% and it was trivial to calculate the neutral response for every record. In the case of specified valence and the reaction based on observed expressions (Emotion Valence) a trivial rate of dependence was shown and no statistically significant differences were demonstrated (Table 5).

Based on the results (Table 5), it was not possible to properly recognize the emotional expressions of the face or the other seven recorded emotions using Affdex software. Comparison of these emotions with specified valence showed a match in only 18.12% of cases. Compared to the recorded observed valence, the expected classification of the examined images by the students even decreased. As it can be seen from the results obtained (Table 5), up to 94% of all the evaluated reactions were neutral. No significant shares for positive and negative emotions (positive emotions = 3%, negative emotions = 2%) were recorded. Based on these results, it could be argued that students did not show enough emotions during the experiment, and therefore Affdex software evaluated their response as a neutral emotion.

## 5. Discussion

In the experiment, it was assumed that the positive (or negative) images evoke positive (or negative) face reactions, as psychologists understand emotions as conscious feelings relating to relevant events, external or internal environment and are associated with specific physiological activation [23,91,92].

Emotions are not only inseparable from decision making, but even the decision making is also an emotional process, because it can trigger a multitude of (mostly negative) emotions [93].

It can be said, that overall, Affdex can achieve acceptable accuracy for standardized images of instructed vs. natural facial expressions, but performs worse for more natural facial expressions. 

Currently applied methods and techniques for automated analysis of facial expressions generate probability-like measures for different basic emotions and therefore, the classifiers are trained with databases of prototypical facial expressions. For this reason, these tools reach high success rates of the overall classification when used on prototypical facial expressions [94,95]. 

This problem that in the case of classifying emotional states, is directly related to the current face analysis systems, has recently been reported by several researchers [3,4]. According to Stöckli [3], methods of solutions of individual phases of recognition process and classification by Affdex itself is standardized but can hinder the generalizations of obtained results. Affdex, as well as other current SDKs, classifies prototypical facial expressions that are unusual in real situations. Therefore, accuracy measures can be inflated. Even though the methods and techniques currently applied in individual SDKs, over the last 20 years, have been improved in terms of algorithmic complexity and the success rate of classification grew as well, the issues plaguing these systems since the 1970s have persisted. According to the study conducted by Stöckli [3], problems such as varying camera angles and changing head poses are being solved. Improvements in the analysis of non-posed faces and the sensitivity of measuring subtle changes in facial expressions and the discrimination of more difficult expressions and expression intensity are expected [87,96].

Current SDKs, supporting fully automated facial analysis often neglect to consider cultural and contextual aspects that can be required to correctly classify the expressed emotional states [97,98]. In real life, emotional facial expressions are not instructed nor uniform but reflect various aspects and are a combination of individual basic emotions [99].

It can be agreed with Abramson et al. [4] that emotional states need to be studied in a complex way. According to the authors [4], emotional states, such as anger, manifest themselves in a person’s brain. Therefore, one of the methods to solve the issues of the current face analysis system is the use of multiple techniques at the same time.

Over the last two decades, special attention has been paid to the automatic recognition of emotions through the extracted areas of the human face. However, there is a growing demand for biometric systems and human-machine interaction where facial recognition and emotional intensity play a vital role [100].

Various biometric sensors (GSR, HR, EEG, and others) can reveal a specific aspect of human cognition, emotion, and behavior. Depending on the research question, it can be therefore considered to combine face analysis using webcam and eye tracking sensors with two or more additional biosensors to gain meaningful insights into the dynamics of attention, emotion, and motivation.

## 6. Conclusions

The results of the experiment show a low success rate of recognition of students’ emotions. Students were shown 100 images that were randomly selected from the standardized database NAPS. The results can be interpreted in two ways. The low success rate may have been caused by improper inclusion of images into the expected reactions. The specified valence of images may have been skewed by the subjective reaction of the expert. That may have caused the deviation, which resulted in low concordance when comparing students’ reactions with expected reaction on a given image. To mitigate the expert’s subjectivity, it can be recommended to use multiple experts from various areas of expertise (psychology, pedagogy, etc.) to set the specified valence. By averaging these values, it would be possible to achieve an optimal specified valence, which would not have the issue of the subjective influence of an expert.

The second interpretation is that the cause of the low success rate may have been caused by insufficient reactions of students on the given images. The students may have not expressed their emotions explicitly enough and it is possible that the Affdex software evaluated their emotional response as neutral. A solution to this problem might be the use of multiple biometric sensors, which could help correctly evaluate the emotional response of participants.

Future work could be focused on optimizing specified valence determination using multiple experts and involving multiple biometric sensors in the process of capturing participant’s responses. The expected result is an increase in the overall success rate of emotion and reaction recognition. Consequently, it would be possible to more accurately analyze students’ responses during the classes, which could lead to optimization of the study content.

The future experiments would contain a subjective evaluation of the images of the participants. During this evaluation, it would be possible to measure the reaction time for activating the appropriate button that would represent a positive or a negative emotional state to the shown image. Neutral emotional status would be recorded if the participant would not push any button. Experiments would be based on the psychological aspect, as images evoke positive or negative emotional states in humans [85]. It can be assumed that the subjective image evaluation (using the button) would correlate with the specified image rating similar to this realized experiment. It would be interesting to measure the reaction time to analyze whether there is a difference in time when deciding between a positive and a negative reaction in response to an emotional state.

## Figures and Tables

**Figure 1 sensors-19-02140-f001:**
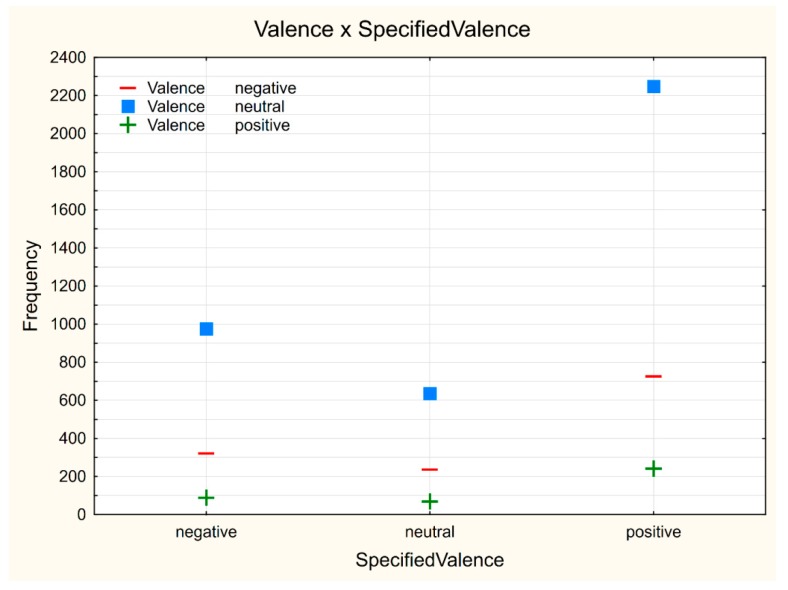
Interaction plot: Valence vs. SpecifiedValence.

**Figure 2 sensors-19-02140-f002:**
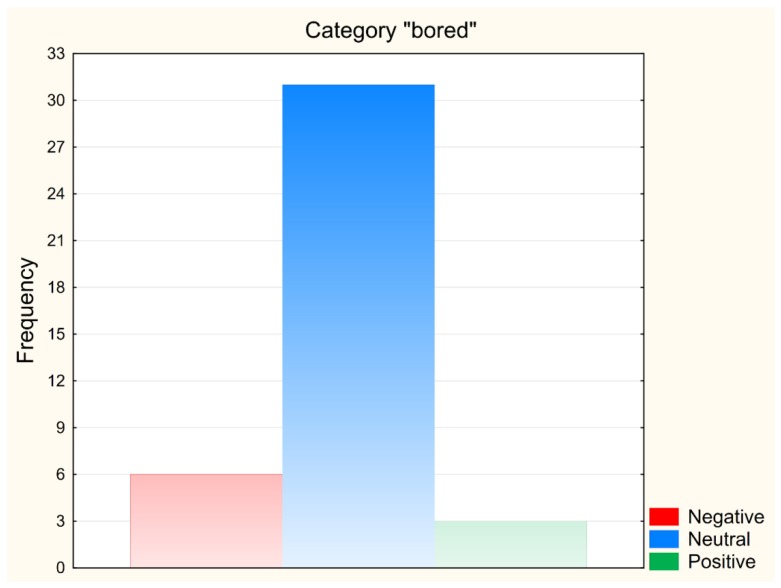
The frequency of recorded reactions for the neutral category “bored”.

**Figure 3 sensors-19-02140-f003:**
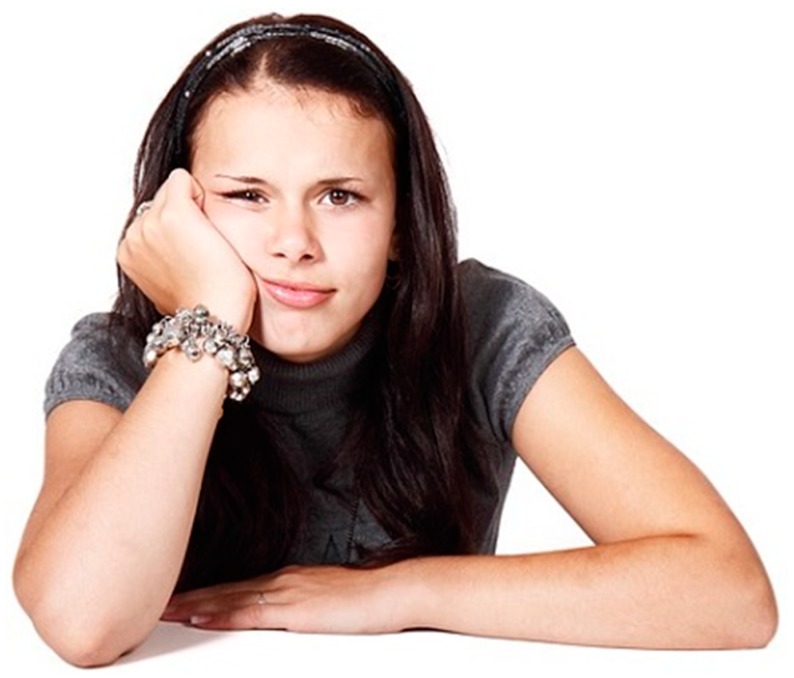
Illustration of the image representing the “bored” category.

**Table 1 sensors-19-02140-t001:** Crosstabulations: Valence x Specified Valence.

Valence\Specified Valence	Negative	Neutral	Positive
**Negative**	322	238	727
	23.23%	25.21%	22.60%
**Neutral**	975	636	2248
	70.35%	67.37%	69.88%
**Positive**	89	70	242
	6.42%	7.42%	7.52%
**∑**	1386	944	3217
	100%	100%	100%
**Pearson**	*Chi-square* = 4.647288; *df* = 4; *p* = 0.32544		
**Con. Coef. C**	0.0289327		
**Cramér’s V**	0.0204671		

**Table 2 sensors-19-02140-t002:** Degree of dependence for negative specified valence.

Negative Specified Valence	
**Pearson**	*Chi-square* = 84.17117; *df* = 66; *p* = 0.06515
**Con. Coef. C**	0.2392752
**Cramér’s V**	0.1742549

**Table 3 sensors-19-02140-t003:** Degree of dependence for neutral specified valence.

Neutral Specified Valence	
**Pearson**	*Chi-square* = 68.07593; *df* = 50; *p* = 0.04537
**Con. Coef. C**	0.2593524
**Cramér’s V**	0.1898872

**Table 4 sensors-19-02140-t004:** Degree of dependence for positive specified valence.

Positive Specified Valence	
**Pearson**	*Chi-square* = 105.8141; *df* = 102; *p* = 0.37815
**Con. Coef. C**	0.1784509
**Cramér’s V**	0.1282423

**Table 5 sensors-19-02140-t005:** Crosstabulations: Emotion Valence x Specified Valence.

Emotion Valence\Specified Valence	Negative	Neutral	Positive
**Negative**	39	37	105
	2.81%	3.92%	3.26%
**Neutral**	1316	886	3032
	94.95%	93.86%	94.25%
**Positive**	31	21	80
	2.24%	2.22%	2.49%
**∑**	1386	944	3217
	100%	100%	100%
**Pearson**	*Chi-square* = 2.554037; *df* = 4; *p* = 0.63499		
**Con. Coef. C**	0.0214528		
**Cramér’s V**	0.0151729		

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
