# Peer review of "A Case Study of Facial Emotion Classification Using Affdex"

_sensors, 2019, doi:10.3390/s19092140_

Round 1

Reviewer 1 Report

In the paper the authors try to find the relation between the valence from images and the expression from observers' face. I think there are some relation between them, but they are different. As mentioned in the paper golden standards are needed for the valence and expression/emotion. The topic is interesting but the paper should be improved greatly.

1. The title of the paper is "Analysis of the reactions captured by the face analysis system Affdex". It’s better to be more specific than using "analysis of the reactions" because the readers cannot know what is the specific topic. "face analysis" is a wide topic including emotion classification, gender classification, age prediction, ..., what ever anything related to face. The authors can use "facial emotion classification".

2. In the second section, the works introduced are mostly now new enough. In recent years the CNN based facial emotion classification achieved a great progress. Besides, there are two many lists in this section. I think this section can be rewritten and reorganized totally.

3. A lot of grammar errors. Many sentences lack the subject. Such as this one. "In this paper is presented an experiment carried out using..." The readers will be confused by it.

4. In the experimental description, it is emphasized that the students are from the course Logic Systems and the course is difficult. But is there any relation between the content of the course and the experiments? I think the description isn’t related to the experiments and can be removed.

Author Response

Dear Reviewer,

Thank you for the opportunity to revisit our paper “Analysis of the reactions captured by the face analysis system Affdex” and fix the critical issues.

We appreciate your comments which helped us to improve our paper.

Please, find below our response to the your comments.

Reviewer comments:

The title of the paper is "Analysis of the reactions captured by the face analysis system Affdex". It’s better to be more specific than using "analysis of the reactions" because the readers cannot know what is the specific topic. "face analysis" is a wide topic including emotion classification, gender classification, age prediction, ..., whatever anything related to face. The authors can use "facial emotion classification".

Authors respond:

If it is possible for us to change the title then we would change it to “Facial emotion classification of reactions using Affdex”.

Reviewer comments:

In the second section, the works introduced are mostly now new enough. In recent years the CNN based facial emotion classification achieved a great progress. Besides, there are two many lists in this section. I think this section can be rewritten and reorganized totally.

Authors respond:

We revisited the section 2 (Related Work) and added another new research.

Reviewer comments:

A lot of grammar errors. Many sentences lack the subject. Such as this one. "In this paper is presented an experiment carried out using..." The readers will be confused by it.

Authors respond:

We revisited the grammar issues of the whole paper and improved the readability.

Reviewer comments:

In the experimental description, it is emphasized that the students are from the course Logic Systems and the course is difficult. But is there any relation between the content of the course and the experiments? I think the description isn’t related to the experiments and can be removed.

Authors respond:

We revisited the section 4 (Results) where we removed the description of the course because as you stated it is not relevant to this experiment.

Reviewer 2 Report

The paper investigated an interesting issue: the reliability of facial expression recognition systems. It focused on one particular system: Affdex. Through exposing the weakness of the Affdex system, the authors claim that the available emotional recognition systems through facial expression recognition are not reliable and may be effective in classifying students' emotions.

The research, however, has its issues itself. First, it only investigates one system. Using the result of studying one facial expression recognition system to generalize is a bit risky. Secondly, one may have questions on the experiment procedure. As the authors stated, showing the images to the students, expecting them to react sufficiently different from one image to another many not work, regardless which facial expression system is tested. Most students of age 18 or 19 probably do not think many of those images funny, disgusting or otherwise. Therefore any subseuqent experiments may not be meaningful. Third, adding more sensors to measure people's emotion states is a trivial solution, which means a solution anyone could think of.

The paper is poorly written, therefore needs to be thoroughly edited. Listed below are some of the sentences which require editing:

The models researched in the field of emotion recognition [13–16], there 63 are two that have been predominantly used in the last decade:…

Due to this reason, can be considered various other 75 frequently used approaches [21,22],…

However, the three outlined approaches are the most widely used and from the 85 historical perspective, most research activity has been devoted to the three approaches.

Application of Gabor filters outputs coefficients that…

Because of that was chosen the Affdex module for the experiment…

Also, the Affdex module offers the availability of the license and its application for testing in the academic environment…

After preparing the data, the log file consisted of 5547 records.

Each record representing a recorded student’s response for a specific image.

At first, the recorded response of the emotional state for each image with the specified (expected) image response was compared.

meaning images with expected positive reaction caused a neutral or negative reaction (or vice versa).

This might have been caused either by the wrong classification of displayed images (meaning images with expected positive reaction caused a neutral or negative reaction) or by wrong or inaccurate evaluation of emotion expressed by a student.

…the only reliable method to eliminate the issues of current face analysis system is…

Author Response

Dear Reviewer,

Thank you for the opportunity to revisit our paper “Analysis of the reactions captured by the face analysis system Affdex” and fix the critical issues.

We appreciate your comments which helped us to improve our paper.

Please, find below our response to the your comments.

Reviewer comments:

The research, however, has its issues itself. First, it only investigates one system. Using the result of studying one facial expression recognition system to generalize is a bit risky.

Authors respond:

Based on results of Rizvi et al., the aim of our research was to verify the reliability of the most commonly used SDK- Affectiva's Affdex SDK. Other frequently used face analysis systems operate on a similar principle (eg. FACET, Eyeris EmoVu, InSight SDK). Because of that was chosen the Affdex module for the experiment. Also, the Affdex module offers the availability of the license and its application for testing in the academic environment.

Reviewer comments:

Secondly, one may have questions on the experiment procedure. As the authors stated, showing the images to the students, expecting them to react sufficiently different from one image to another many not work, regardless which facial expression system is tested. Most students of age 18 or 19 probably do not think many of those images funny, disgusting or otherwise. Therefore any subseuqent experiments may not be meaningful.

Authors respond:

Since we use Affdex in a university environment the test sample was represented by students with this age. In the future research we plan to use a broader age sample that could show potential differences based on age, gender, etc.

Reviewer comments:

Third, adding more sensors to measure people's emotion states is a trivial solution, which means a solution anyone could think of.

Authors respond:

We revisited the section 6 (Conclusions) where we improved the description of our future solution.

Reviewer comments:

The paper is poorly written, therefore needs to be thoroughly edited.

Authors respond:

We revisited the grammar issues of the whole paper and improved the readability.

Reviewer 3 Report

This paper shows the result of face recognition system for especially facial expression.

Even though a part of survey on previous systems is well-written, it has some limitation.

While traditional approaches are introduced and summarized very well, recent algorithms are not mentioned.

Recently, deep-learning-based algorithms are usually used. they proposed similar problems with this paper on exaggerated database and suggest to make good database. 

In view of this point, this paper shows good comparison and practical analysis. However, this paper does not have technical contribution and does not included recent approaches. According to references, there are little recent papers on related issues. 

Author Response

Dear Reviewer,

Thank you for the opportunity to revisit our paper “Analysis of the reactions captured by the face analysis system Affdex” and fix the critical issues.

We appreciate your comments which helped us to improve our paper.

Please, find below our response to the your comments.

Reviewer comments:

Even though a part of survey on previous systems is well-written, it has some limitation. While traditional approaches are introduced and summarized very well, recent algorithms are not mentioned. Recently, deep-learning-based algorithms are usually used. They proposed similar problems with this paper on exaggerated database and suggest to make good database. In view of this point, this paper shows good comparison and practical analysis. However, this paper does not have technical contribution and does not included recent approaches. According to references, there are little recent papers on related issues.

Authors respond:

We revisited the section 2 (Related Work) and added another new research dealing with deep-learning-based algorithms.

Round 2

Reviewer 1 Report

The topic is interesting, and this revision has been improved according to the reviewers' comments and suggestions. But I suggest the authors still to check the grammar errors, such as in Section 5, "In the experiment was assumed that the positive ..." There isn't a subject. Besides, there are a lot of lists in the paper. Such as the list following this sentence "In 2002, Yang has introduced a classification [30] that is mostly used by many other authors:". Normally the authors can rewrite the list into a sentence. 

Author Response

Dear reviewer,

Thank you for the opportunity to revisit our paper “Analysis of the reactions captured by the face analysis system Affdex” and fix the critical issues. We appreciate your comments which helped us to improve our paper.

Reviewer comments:

The topic is interesting, and this revision has been improved according to the reviewers' comments and suggestions. But I suggest the authors still to check the grammar errors, such as in Section 5, "In the experiment was assumed that the positive ..." There isn't a subject. Besides, there are a lot of lists in the paper. Such as the list following this sentence "In 2002, Yang has introduced a classification [30] that is mostly used by many other authors:". Normally the authors can rewrite the list into a sentence.

Authors respond:

We have revisited the whole paper and improved the language of the paper. We have also rewritten some of the lists into sentences.

Reviewer 2 Report

The paper is long on overview of the emotion recognition literature, short on the main problem it addresses. However, on the whole, it is still a good paper from which readers can obtain useful information.

I suggest that the title on of the paper title is changed to "A Case Study of Facial Emotion Classification using Affdex." I also suggest that the authors revise the abstract to include "Affdex" somewhere. 

The addition on the advancement of using deep learning for emotion recognition is a welcome addition. 

The revised version of the paper is significantly better in presentation than the original version. However, the sentence structures of a few sentences are still not correct. For instance, in English,  we say, "The apple is red," not "Is red the apple," or "Is apple red." Please find these instances and correct them.

Author Response

Dear reviewer,

Thank you for the opportunity to revisit our paper “Analysis of the reactions captured by the face analysis system Affdex” and fix the critical issues. We appreciate your comments which helped us to improve our paper.

Reviewer comments:

The paper is long on overview of the emotion recognition literature, short on the main problem it addresses. However, on the whole, it is still a good paper from which readers can obtain useful information.

I suggest that the title on of the paper title is changed to "A Case Study of Facial Emotion Classification using Affdex." I also suggest that the authors revise the abstract to include "Affdex" somewhere.

The addition on the advancement of using deep learning for emotion recognition is a welcome addition.

The revised version of the paper is significantly better in presentation than the original version. However, the sentence structures of a few sentences are still not correct. For instance, in English,  we say, "The apple is red," not "Is red the apple," or "Is apple red." Please find these instances and correct them.

Authors respond:

We have revisited the whole paper and improved the language of the paper. If it is possible we would like to change the title to the proposed one.

Reviewer 3 Report

Even though this paper included some papers on deep learning, some important papers are missing. Because this paper deals with comparisons between computer system and human being, it should include recent works. However, In this paper, some key papers are missing such as;

"Deep Facial Expression Recognition: A Survey," Shan Li and Weihong Deng, arXiv:1804.08348v2 [cs.CV] 22 Oct 2018

"A Compact Deep Learning Model for Robust Facial Expression Recognition," CVPR 2018

"Deep learning for facial expression recognition: A step closer to a smartphone that knows your moods," CVPR 2017

Author Response

Dear reviewer,

Thank you for the opportunity to revisit our paper “Analysis of the reactions captured by the face analysis system Affdex” and fix the critical issues. We appreciate your comments which helped us to improve our paper.

Reviewer comments:

Even though this paper included some papers on deep learning, some important papers are missing. Because this paper deals with comparisons between computer system and human being, it should include recent works. However, In this paper, some key papers are missing such as;

"Deep Facial Expression Recognition: A Survey," Shan Li and Weihong Deng, arXiv:1804.08348v2 [cs.CV] 22 Oct 2018

"A Compact Deep Learning Model for Robust Facial Expression Recognition," CVPR 2018

"Deep learning for facial expression recognition: A step closer to a smartphone that knows your moods," CVPR 2017

Authors respond:

Thank you for the recommendation, we have added the recommended articles to our paper. We have also improved the language in the paper.